# Changes in Richness and Species Composition after Five Years of Grazing Exclusion in an Endemic Pasture of Northern Mexico

José Ramón Arévalo [1], Cristina González-Montelongo [1], Juan A. Encina-Domínguez [2],*, Eduardo García [3] and Miguel Mellado [3]

1. Department of Botany, Ecology and Plant Physiology, University of La Laguna, Islas Canarias, 38200 La Laguna, Spain
2. Department of Natural Resources, Autonomous Agrarian University Antonio Narro, Saltillo 25315, Mexico
3. Departament of Animal Nutrition, Autonomous Agrarian University Antonio Narro, Saltillo 25315, Mexico
* Correspondence: juan.encinad@uaaan.edu.mx

**Abstract:** A well-managed grazing system improves the productivity and health, and it is important to promote sustainability. We analyzed the impact of grazing on the Sierra de Zapalinamé protected area in north Mexico. Our hypothesis was that grazing modifies species composition, richness, and nutrients after grazing exclusion for five years. In this area, eight plots were excluded from grazing, and species richness, evenness, and plant functional types for five years were monitored. This monitoring was also carried out on eight control plots adjacent to the excluded plots. Soil samples were collected from each plot in the fifth year of exclusion for nutrient content analysis. Grazing discriminated plant species composition after five years between excluded and control plots, but not species richness and evenness. In addition, exclusion increased grass cover and decreased forb cover. Indicator species for excluded and control sites were identified. It was concluded that part of the pastures can be excluded from grazing as a way to analyze changes in this protected area and promote greater plant diversity.

**Keywords:** DCA; evenness; grazing; pastures; richness





## 1. Introduction

Grazing constitutes an important ecological force with huge environmental and social implications and must be analyzed thoroughly to develop sustainable practices [1]. Grazing is one of the most important traditional and sustainable land uses in many areas of the world [2,3]. Thus, pasture grazing requires direct management techniques to maintain species composition, soil conservation, and high diversity in plant communities [4,5]. On the contrary, grassland mismanagement can cause obvious and significant variation in species composition [6,7]. Additionally, overgrazing is a common practice in many pastures, increasing soil erosion and desertification and promoting the spreading of exotic species [8–12].

In the North American pastures, a mix of tall and short grasses predominates, which are distributed from southern Canada to central Mexico [13]. The semiarid pasture of Mexico is considered a type of shortgrass prairie distributed from Alberta to Arizona, New Mexico, Texas, and northern Mexico [14]. There are many similarities between the shortgrass and the semiarid Mexican pastures, including the genus *Bouteloua*, a dominant species in these ecosystems [15], Mexico being the center of diversity for this genus, with 29 species and 13 subspecies [16].

In most areas of northern Mexico, rangeland overgrazing by goats, sheep, cattle, or horses is common [17,18]. These authors recommend recovery periods for the biomass. In some cases, some highly palatable plants for cattle, goats, or horses can be promoted, while other shrub species can be negatively affected [18]. Overgrazing in South Africa has given similar results to those observed in Mexico [19]. In short, overgrazing results in a steady

decline in the condition of pastures, evidenced by a reduction in palatable forage plants and in the plant species composition [9]. The final consequence of overgrazing is infertile soil and an aboveground biomass reduction as well as an overall pasture productivity decline [20]. Perceptual evidence of changes in soil and vegetation patterns and socio-economic issues (such as land tenure and forms of organization) are now factors that have to be considered for rangeland management [21].

In this study, the effects of horse–cattle grazing on the structure of northern Mexican pastures were determined. In addition, it was determined whether such effects are significant in modifying species composition, richness, or evenness after five years of grazing exclusion.

The hypothesis was that cattle grazing is an important environmental determinant of plant composition, species richness, and evenness in northern Mexico pastures. As long as grazing effects determining species composition, richness, and evenness can be controlled by management [5], this information can be valuable for managers in the decision-making procedures for arid and semiarid ecosystem conservation.

## 2. Materials and Methods

### 2.1. Study Site

The study site is located in southeastern Coahuila State, a transition area between the Chihuahuan desert and the Oriental Sierra Madre physiographic province ($25°13'57.48''$–$25°14'57.25''$ N and $100°56'44.62''$–$101°01'5.17''$ W). This area lies within the natural protected area of Sierra de Zapalinamé (Figure 1).

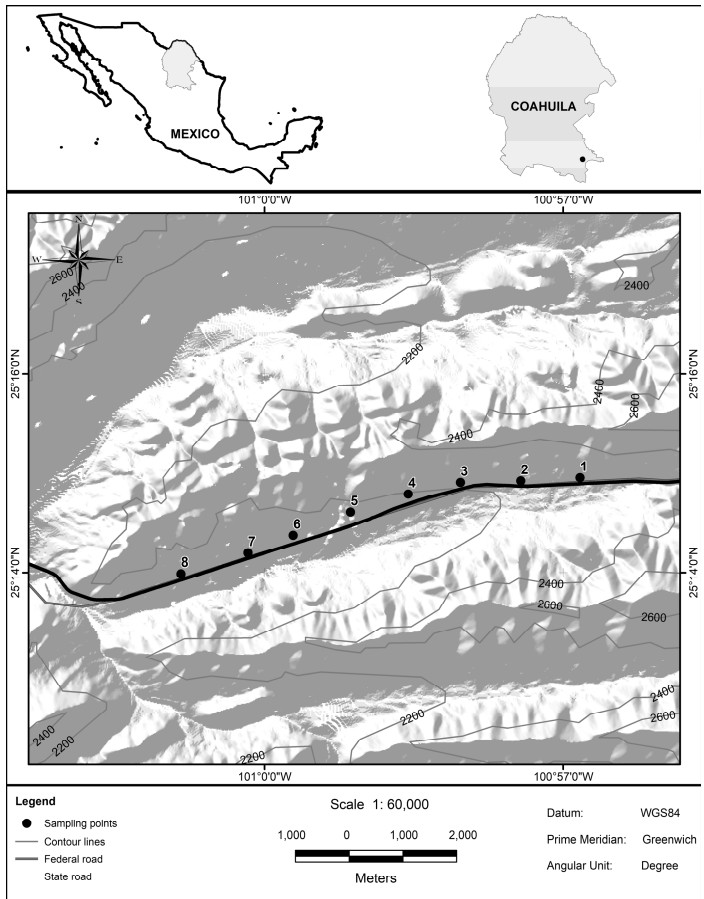

**Figure 1.** Study site showing the sampling plots located along the pastures analyzed and the location in the Coahuila State, Mexico.

The plots were established at an altitudinal range of 2102 to 2268 m.a.s.l. Climate conditions in the region are dry and are classified as BSKw semiarid templated weather with dominant precipitation during summer [22]. The site presents dominant calcarean rocks and deep well-drained soils. The average annual temperature is 16.9 °C and the average annual precipitation is 498 mm. Plant community is dominated by species of the genus *Bouteloua*, such as *B. dactyloides*, *B. gracilis*, *B. uniflora*, and *B. curtipendula*, and with the presence of *Aristida havardii*, *A. pansa*, and *Muhlenbergia phleoides* [23]. Predominant woody species are *Buddleja scordioides*, *Gymnosperma glutinosum*, *Mimosa biuncifera*, and *Prosopis glandulosa*.

Agriculture activity was initiated in this area at the end of the XIX century [24] with wheat, corn, beans, and barley as main crops. Also, some fruit trees were cultivated in the pasture areas and alluvial valleys.

At the present time, pastures in the study site (approximately 400 ha) are grazed by cows and horses. The number of livestock is relatively constant, with 63 cattle heads and 37 equines.

### 2.2. Design of the Experiment

In March 2017, in the center of the stand of the main pasture community in the Sierra of Zapalinamé natural protected area, we systematically located, along a transect, eight square pairs of plots ($20 \times 20$ m$^2$), separated approximately 1000 m from each other. From these pairs of plots, one of them was excluded from grazing. In the center of each plot (control and excluded), we concentrically established a $10 \times 10$ m$^2$ permanent plot. Data were from the latter plots ($10 \times 10$ m$^2$), which were the ones sampled. The pairs of plots, control and excluded, were separated by a minimum of 10 m.

In each plot, altitude, aspect, and slope were measured. We also visually estimated the percentage of rock, bare soil, litter cover, grass cover, and understory woody species cover. We identified all herbs and shrubs in the plot. Cover for all the species on plot surfaces was visually estimated and noted on a scale from 1 to 9, according to the following cover classes: (1, traces; 2, >1% cover in the plot; 3, 1–2%; 4, 2–5%; 5, 5–10%; 6, 10–25%; 7, 25–50%; 8, 50–75%; and 9, >75%). Samplings were carried out from 2017 to 2021 in August (the humid period of the year). Rainfall was relatively high in 2016 (over 500 mm) and decreased over the years to approximately 200 mm. We can assume that after a humid year, the rest of the years were dryer than average. However, average annual temperature remained relatively constant, with a variation of less than 0.5 °C (Figure 2).

Taxonomic identities of collected plant specimens were determined, and vouchers were deposited at the ANSM herbarium (Autonomous Agraria University of Antonio Narro Herbarium). For species names, we followed the checklist of vascular plants of the Sierra of Zapalinamé [23]. Plot position and elevation were measured using a global positioning system (GPS; Etrex, Garmin Ltd., Olathe, KS, USA).

In 2021, four soil samples were collected (from 0 to 10 cm depth), 20 cm out of the corner of each plot. These were mixed, dried, and passed through a 2 mm sieve; debris and stones were eliminated. Organic matter content was determined by the Walkley and Black method [25], and pH was measured in a soil-to-water ratio of 1:5 extract. Soil total nitrogen (TN), extratable phosphorus (using the Olsen method (P Ols)), K, Na, Mg, Ca, Cu, Zn, Fe, Mn, B, and S were determined. We also calculated Cation Exchange Capacity [26,27].

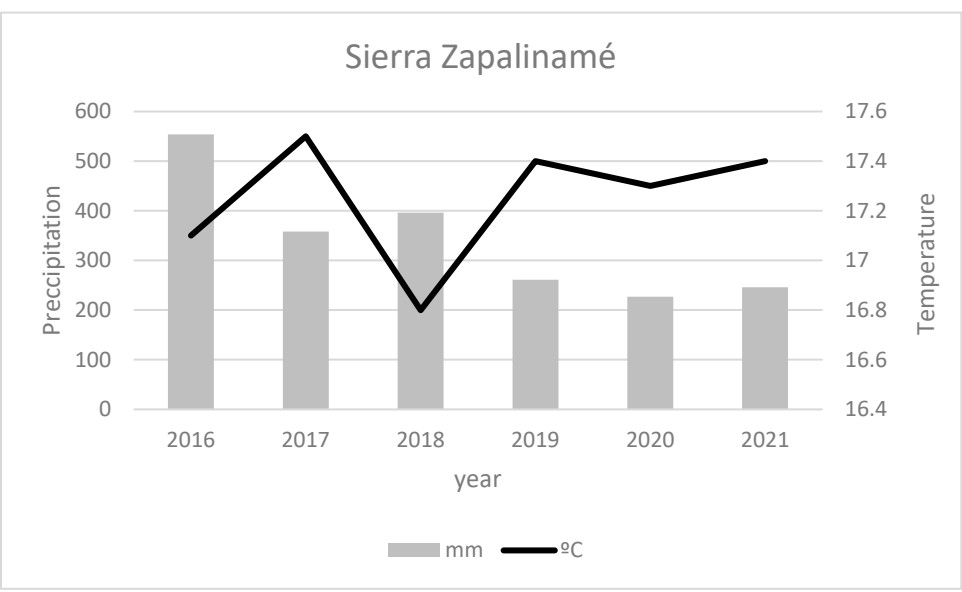

**Figure 2.** Average annual precipitation and average annual temperature in the area of Sierra Zapalinamé throughout the years of the study.

*2.3. Statistical Analyses*

We evaluated the effect of the factors—control vs. exclusion and sampling year—in grass, forbs, and woody cover (used as functional types) of the plots, using the GLM procedure with the main effects as fixed effects and the pairs of plots as random factor. The homogeneity of variances was checked using Bartlett's test (for a $p < 0.05$). The same analysis was used to evaluate the effect of both factors on species richness using the Smith and Wilson Evenness Index [28].

Ordination techniques help to explain community variation [29,30], and they can be used to evaluate trends through time as well as space [31]. We used Principal Components Analysis (PCA) to examine the soil chemical composition in 2021 to analyze differences in nutrient composition, as we expect a linear gradient of nutrient characteristics with respect to sites.

To determine whether PCA axes discriminated species composition between control and excluded plots, the I and II axes scores were analyzed using logistic regression, using the pairs of plots as the covariable matrix to reduce the spatial variability in the analyses, and using the $\chi^2$ statistic to determine its significance in discriminating both plots (control vs. exclusion; for $p < 0.05$).

We used DCA (Detrended Correspondence Analysis; [32]) to analyze the species composition (based on species cover) of control vs. excluded plots, using also the pairs of plots as the covariable matrix. As expected, species distributed unimodally through environmental gradients, as was assumed by DCA. The CANOCO program version 5.1 (Microcomputer Power, Ithaca, NY, USA) was used for all multivariate analyses [31]. Again, to determine axes I and II, coordinates were analyzed with GLM logistic regression.

An MRPP (Multi-Response Permutation Procedure) was used to determine changes in species composition between control and exclusion plots with a matrix base in cover. The Bray–Curtis distance was used for this analysis [33]. For the same data matrix, an Indicator Species Index (ISI) was used to determine the significant representative species in each group [34]. The analyses were carried out in the Vegan R Package [35].

**3. Results**

Environmental characteristics in the study area under the same management were relatively constant along the transect: altitudinal variation of less than 150 m, same aspect and slope (from 10 to 20 sexagesimal degrees), and relatively constant grazing pressure (Table 1).

**Table 1.** Plot characteristics and nutrient content of the plots ("E" for exclusion plots and "C" for control plots). Aspect measured in degrees and slope in sexagesimal degrees. Nutrient content is measure in percentage (%) or mg/kg.

| | | | | | % | | | | | | | mg/kg | | | | | | |
|---|---|---|---|---|---|---|---|---|---|---|---|---|---|---|---|---|---|---|
| Plot | Alt (m a.s.l.) | Aspect | Slope | pH | OM | P Ols | K | Ca | Mg | Na | TN | Fe | Zn | Mn | Cu | B | S |
| E1 | 2243 | 140 | 15 | 7.99 | 7.04 | 15.9 | 519 | 4147 | 136 | 15.5 | 21.0 | 4.3 | 0.21 | 8.27 | 0.6 | 1.95 | 4.22 |
| C1 | 2246 | 140 | 15 | 8.10 | 7.02 | 16.5 | 489 | 4440 | 144 | 14.5 | 12.1 | 3.59 | 0.29 | 7.96 | 0.58 | 1.8 | 2.81 |
| E2 | 2231 | 140 | 23 | 8.00 | 6.22 | 18.7 | 315 | 4418 | 134 | 10.0 | 12.8 | 4.42 | 0.25 | 7.6 | 0.58 | 1.71 | 1.41 |
| C2 | 2235 | 140 | 23 | 8.16 | 4.54 | 18.3 | 330 | 4287 | 142 | 11.5 | 10.7 | 4.02 | 0.28 | 8.37 | 0.59 | 1.71 | 7.03 |
| E3 | 2213 | 140 | 20 | 8.09 | 7.30 | 20.9 | 274 | 4230 | 142 | 16.6 | 12.5 | 4.04 | 0.35 | 9.32 | 0.5 | 1.95 | 1.41 |
| C3 | 2220 | 140 | 20 | 8.09 | 7.19 | 20.1 | 498 | 4294 | 163 | 12.6 | 12.8 | 3.25 | 0.25 | 7.68 | 0.57 | 1.93 | 1.41 |
| E4 | 2194 | 135 | 25 | 8.09 | 6.57 | 28.4 | 296 | 3927 | 149 | 17.0 | 13.6 | 3.72 | 0.25 | 6.89 | 0.58 | 1.92 | 1.41 |
| C4 | 2198 | 135 | 25 | 8.01 | 6.66 | 17.0 | 245 | 3934 | 136 | 14.2 | 12.4 | 4.63 | 0.24 | 7.6 | 0.55 | 1.85 | 1.41 |
| E5 | 2171 | 140 | 18 | 7.95 | 5.32 | 20.4 | 186 | 3689 | 178 | 17.5 | 12.8 | 4.44 | 0.27 | 7.2 | 0.4 | 1.93 | 1.41 |
| C5 | 2180 | 140 | 18 | 8.17 | 5.43 | 18.7 | 183 | 3675 | 200 | 12.4 | 12.2 | 3.7 | 0.28 | 7.27 | 0.43 | 1.79 | 1.41 |
| E6 | 2131 | 140 | 12 | 8.09 | 6.18 | 16.9 | 284 | 3938 | 159 | 11.8 | 15.0 | 3.16 | 0.23 | 7.51 | 0.45 | 2.02 | 1.41 |
| C6 | 2138 | 140 | 12 | 8.11 | 6.89 | 16.9 | 288 | 4081 | 160 | 16.1 | 15.7 | 3.16 | 0.2 | 8.06 | 0.45 | 1.98 | 5.63 |
| E7 | 2129 | 140 | 10 | 8.28 | 3.13 | 13.8 | 233 | 3328 | 367 | 7.05 | 8.21 | 3.3 | 0.36 | 5.84 | 0.43 | 1.42 | 5.63 |
| C7 | 2125 | 140 | 10 | 8.16 | 3.15 | 15.1 | 210 | 3195 | 357 | 12.8 | 8.62 | 3.24 | 0.24 | 5.94 | 0.43 | 1.51 | 1.39 |
| E8 | 2112 | 140 | 10 | 8.16 | 5.89 | 23.5 | 147 | 3847 | 249 | 19.1 | 10.4 | 3.27 | 0.29 | 8.87 | 0.4 | 1.81 | 1.41 |
| C8 | 2115 | 140 | 10 | 8.11 | 6.79 | 27.1 | 222 | 3906 | 238 | 15.3 | 12.5 | 3.42 | 0.28 | 8.93 | 0.42 | 1.87 | 1.41 |

Regarding grass, forbs, and woody cover compared between treatments and considering the effect of year, the GLM model (with the pairs of plots as a random factor) showed significant differences, with higher values for grass cover in the excluded plots vs. control plots, but not for the year or factors interactions ($F_{1,75} = 8.75$, $p < 0.01$ for the treatment factor; Figure 3a). In the case of forbs, the same result was found with a significant year and treatment effect ($F_{1,75} = 4.89$, $p < 0.05$ for the treatment factor; Figure 3b), but in this case, the values were higher in the control plots. On the other hand, for woody plants, we obtained a different result, with a significant effect of year ($F_{1,75} = 12.03$, $p < 0.01$; Figure 3c) and non-significant effect for treatment, revealing the same growth of woody plants in both treatments along the years. For all cases, the Bartlett test did not show significant heteroscedasticity ($p > 0.05$).

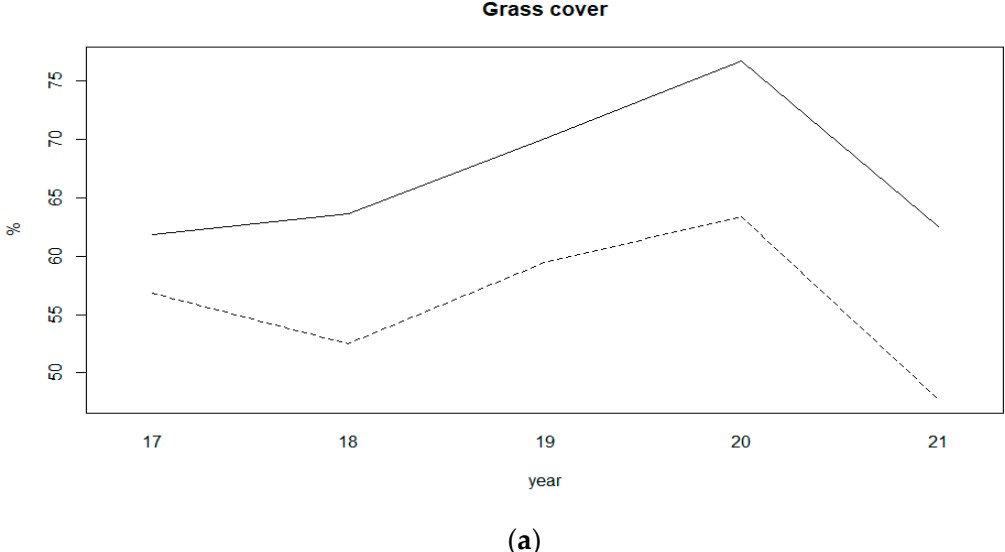

(a)

**Figure 3.** *Cont*.

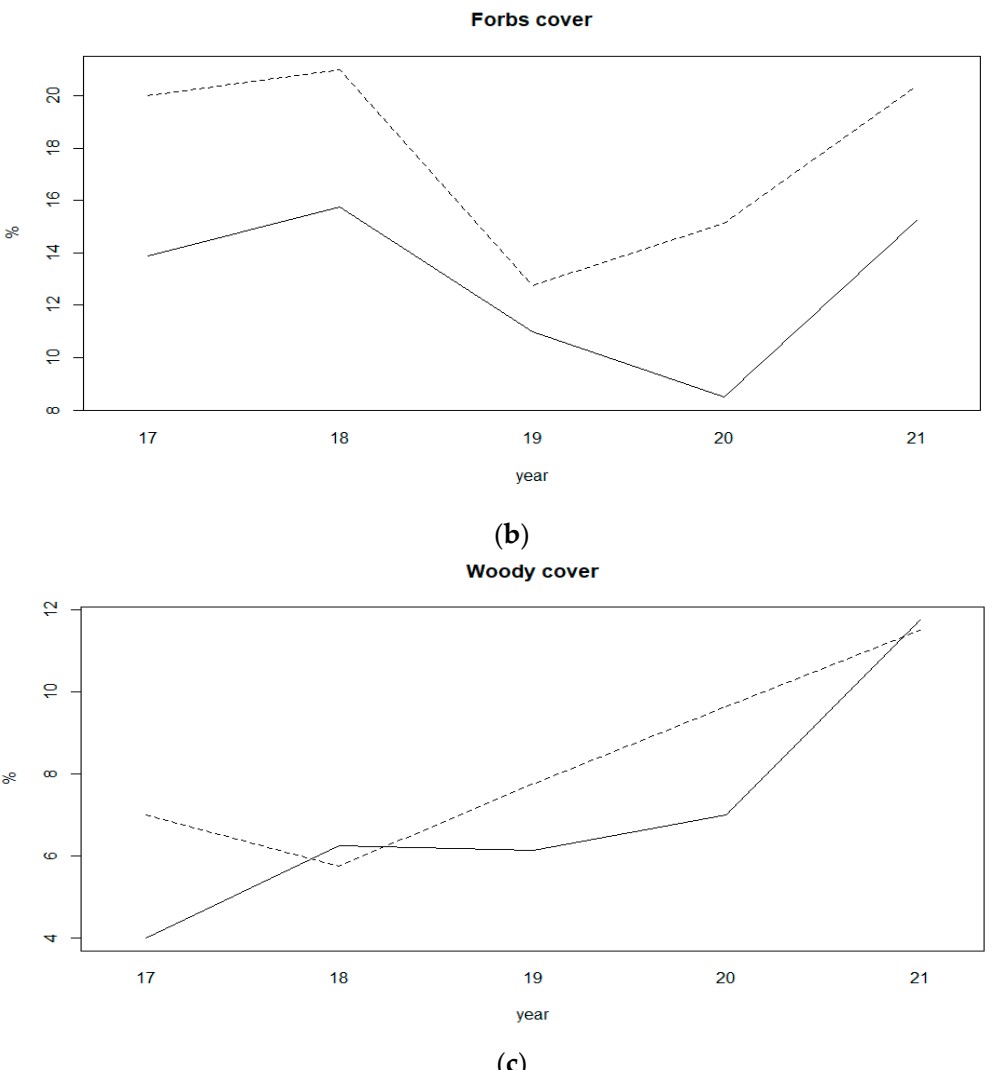

**(c)**

**Figure 3.** Average percentage variation of (**a**) grass cover, (**b**) forb cover, and (**c**) woody species cover along the five sampling years. Solid line for excluded plots vs. dotted line for control plots.

A total of 161 species were found (Appendix A), of which only three were introduced (*Asphodelus fistulosus*, *Malva parviflora*, and *Tribulus terrestris*), but the analysis did not reveal dominance of these introduced species on the control and excluded plots. Regarding species richness, non-significant differences existed between sites ($F_{1,75} = 0.684$, $p > 0.05$); however, the sampling year affected the presence of these plants ($F_{1,75} = 4.533$, $p < 0.05$), with an increase in both treatments along five years of 4–6 species (interaction among factors was also non-significant $F_{1,75} = 0.504$, $p > 0.05$). In the case of the Smith and Wilson Evenness Index, results revealed differences among years ($F_{1,75} = 18.799$, $p < 0.01$), but not for treatment or the treatment × year interaction (control vs. exclusion; $F_{1,75} = 0.204$, $p < 0.05$, and $F_{1,75} = 0.020$, $p < 0.05$), which decreased by year for both treatments, from 0.90 to 0.85 (Figure 4).

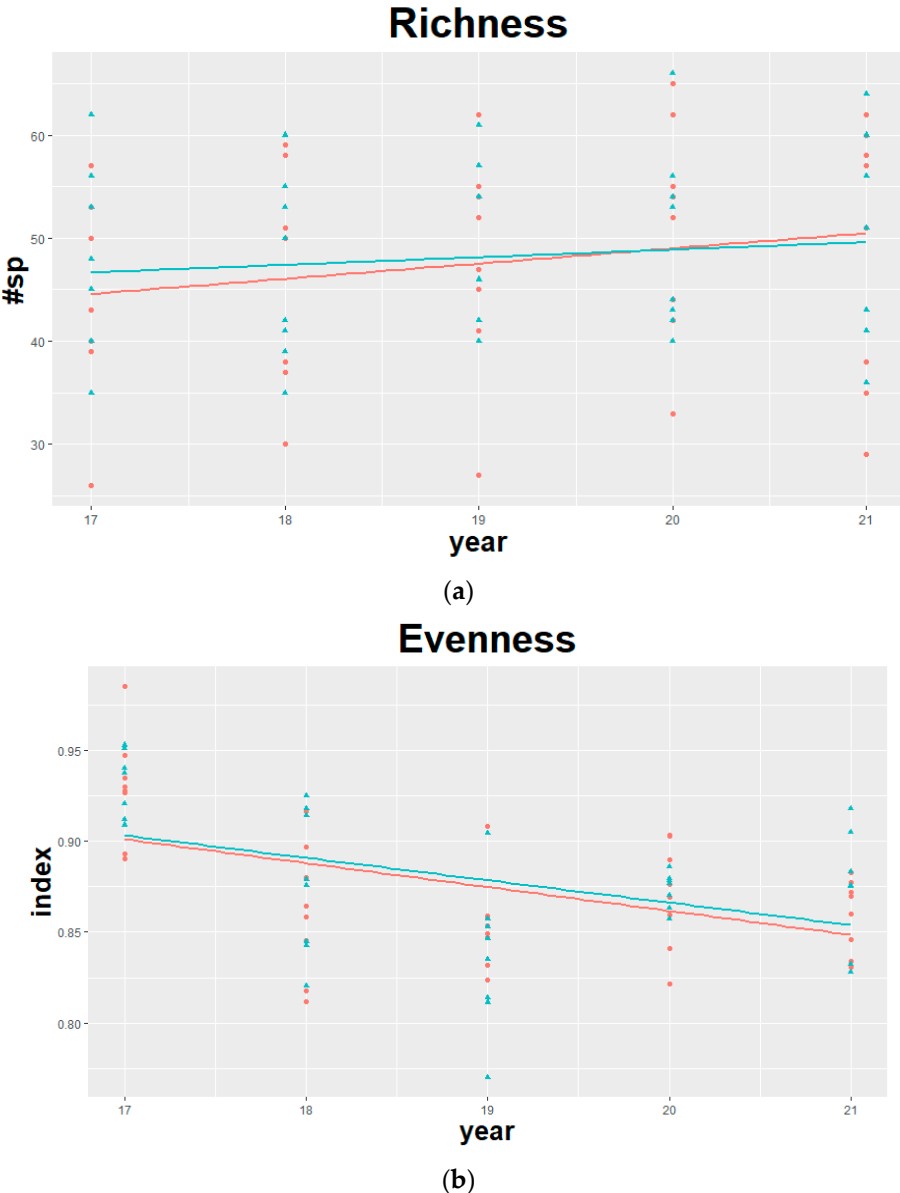

**Figure 4.** (**a**) Linear variation of species richness in ungrazed and grazed plots. Blue lines and dots for control and red for exclusion plots. (**b**) Same presentation of the Smith and Wilson Evenness Index.

The PCA of nutrient composition on the plot pairs concerning the treatment did not reveal discrimination among treatments; although PCA axis II in some way discriminated the polygons, this discrimination was not significant for any of the axes (Z value with the GLM binomial model of 0.037 with $p < 0.05$ and for axis II, Z = 1.27, $p < 0.05$). Although there were some observed tendencies, such as K and Cu or P Olsen, which were more important in the control plot, and Mn and Fe, which were more important in the exclusion plots, these were not different from a nutrient composition perspective (Figure 5).

The ordination analysis for the species cover of plots gave different results. Axis I discriminated control vs. exclusion plots based on GLM binomial analysis of the scores (Z = 2.88, $p < 0.01$), while it was not significant for axis II (Z = 1.72, $p > 0.05$). In the case of the ordination space, *Tribulus terrestris*, *Amaranthus hybridus*, *Sphaeralcea angustifolia*, *Alternanthera repens*, *Urochloa meziana*, or *Phemeranthus aurantiacus* were representative of the control plots, all of which are shrubs or forbs. For the exclusion plots, the dominant species in the ordination space were *Drymaria anomala*, *Bothriochloa barbinodis*, *Phyllanthus polygonoides*, *Bouteloua hirsuta*, *Aristida purpurea*, *Desmanthus painteri*, *Turbinicarpus beguinii*,

*Echeandia flavescens*, *Cyperus niger*, or *Bouteloua curtipendula*, with a dominance of graminoids (Figure 6).

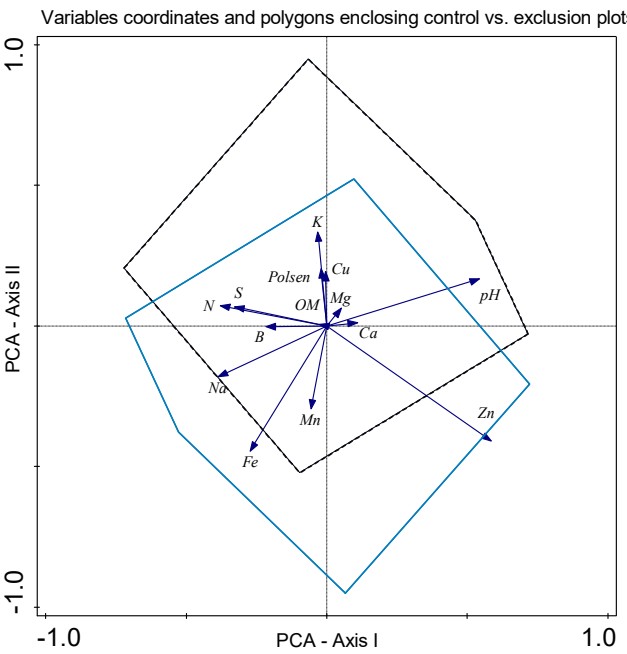

**Figure 5.** Principal components analysis (PCA) of the soil samples of the pairs of plots in 2021 and soil nutrient variables. Variables used appear as arrows, while control plots are enclosed in a black dashed line, and exclusion plots are enclosed in a blue solid line. The matrix used for this analysis appears in appendix II (eigenvalue for axis I: 0.08, eigenvalue for axis II: 0.05, cumulative percentage of variance explained for axes I and II: 63.4%).

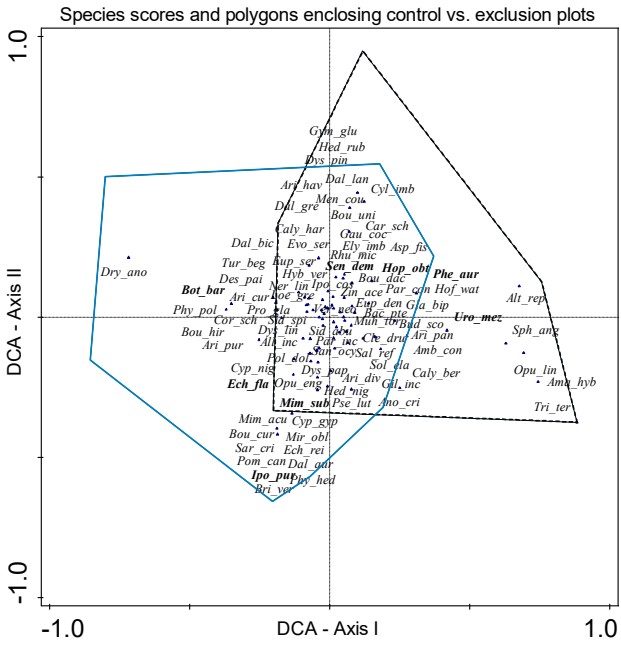

**Figure 6.** Detrended correspondence analysis axes I and II. Species coordinates and control vs. exclusion plots coordinates. Polygons enclose plots under the same treatment: polygon with a dashed black line for control plots and solid blue line for exclusion plots (species names use the three first letters of the genus, followed by the three first letters of the specific epithet from Appendix A). Eigenvalues for axis I: 0.06, eigenvalue for axis II: 0.05, the cumulative percentage of total inertia for axes I and II: 15.6%.

Differences between treatments based on species cover were significant (MRPP), with T = −1.59 and group probability correction of A = 0.0043 ($p < 0.05$). The ISI base in 1000 permutations revealed that the species indicator for control plots were *Urochloa meziana*, *Phemeranthus aurantiacus*, *Senna demissa,* and *Hopia obtusa*, while *Ipomoea purpurea*, *Mimosa subinermis*, *Bothriochloa barbinodis*, and *Echeandia flavescens* were indicators for the exclusion plots ($p < 0.05$ for all of them). These indicator species were represented in the DCA biplot remarked in bold letters (Figure 6).

## 4. Discussion

The study was carried out in a natural protected area where appropriate grazing management has been implemented for several decades. Responses of plant communities to cattle and horse grazing exclusion were partly species-specific, with larger values for grasses in the excluded plots and a greater number of forbs in the control plots, whereas woody species varied along the years with no differences between grazed and ungrazed sites. These results are in line with previous studies where grasses have increased in sites excluded from cattle grazing [36–39]. This suggests that this stratum has the ability to spread as a consequence of grazing exclusion. Thus, cattle grazing exclusion can offer appropriate habitat conditions for increasing the abundance of the existing graminoids. Non-legume forbs, on the other hand, increased their relative cover in the grazed area. Forbs constitute approximately 15% of cattle diet in the Chihuahuan Desert Rangelands [40,41]; thus, although forbs are not a major part of cattle's diet in semiarid rangelands, these plants are important particularly in summer and fall because of their high nutrient content [42]. Contrary to the view that cattle grazing reduces standing plant biomass, potentially reducing cover and forage availability, the present study showed that cattle grazing increased some forb densities, which is in line with [43]. In addition, [44] did not find evidence that species richness of forbs responded to 5 years of cattle removal. The higher canopy cover of forbs in the grazed area in the present study could be possibly due to ground disturbance by trampling of cattle, which can benefit forbs (e.g., [45]). Moreover, the grassland community studied seems to be relatively stable and resilient in response to the intensity of cattle grazing currently practiced. Additionally, forbs in this grassland community may be well-adapted to disturbances derived from cattle grazing.

Woody species were not modified by livestock exclusion when considering the variables that were measured. This response could be because the exclusion time evaluated was insufficient to cause measurable changes in woody plants, or grazing pressure was not intense. The same response was observed in dry forest ungrazed during 7–8 years [46]. However, other studies have shown that heavy grazing modifies richness, density, and species diversity of woody plants [47,48], as well as spatial distribution [49], reduction in tree emergence, and survival [50].

A high number of plant species was found in this study, but we did not find differences in species richness along the years between control and ungrazed plots, nor did evenness reveal significant differences among treatments. Some studies measuring grazing effects on plant species composition and species richness have been inconsistent and conflicting in their results, lacking a general model that predicts the response of grazing intensity or abandonment [4,51]. This lack of consistent results has been attributed to high factor variability, such as the evolutionary history of grazing, productivity gradients, or grazing intensity [52]. Several studies have found higher evenness in grazed than ungrazed treatments [53–55]. A metanalysis study has revealed that the evolutionary history of grazing and grazing intensity alone does not significatively explain changes in species richness, suggesting that the intermediate disturbance hypothesis cannot be supported [56]. In our case, we can also consider that exclusion time has been not long enough to reveal differences in richness, but these differences have been found in the coverage analysis considering the plant functional types [57]. Temperature and humidity can explain some of the changes in functional types [58]; we did not find such a relationship, which can be related to the short

sampling period or the low variability of these parameters (except for the high precipitation during 2016).

In the case of nutrient soil composition, after five years, the variation was not significant, as revealed by the PCA analysis that did not discriminate plots of both treatments. Although grazing is an important factor in modifying nutrient soil composition [59,60], in this case, again, the excluded time probably was not enough to produce these changes [61]. This is in agreement with the result that species richness is related to changes in carbon total nitrogen in soil [62,63]. In addition, the lack of difference in soil mineral content in the five years' exclusion can be partially explained by the fact that the grazing intensity was not high enough, and consequently, vegetation cover did not differ much between the grazed and ungrazed plots.

The DCA revealed that grazed vs. exclusion plots were significantly discriminated based on the total species cover. Although species richness did not reveal significant differences, after five years of grazing exclusion, the plant community proved to be different, with some differences in species. However, despite the high palatability of *Hopia obtusa* and *Urochloa meziana*, these grasses were abundant in the grazed plots. In contrast, in the exclusion plot, only one grass (*Bothriochloa barbinodis*) was markedly abundant. Thus, the impact goes further than just functional types, and generalizations are not possible [1,64,65].

In grazed plots, annual weeds [66], such as *Ambrosia confertiflora*, *Euphorbia dentata*, *Sphaeralcea angustifolia*, and *Solanum elaeagnifolium*, also grow. These are dispersed by grazing livestock from croplands adjacent to the grassland. It is important to mention that *Ophioglossum engelmannii*, a native fern rare to this mountain range, was present in two control plots as were another two rare species, *Pomaria canescens* and *Dichromanthus michuacanus* [67]; on the other hand, in the grazed plots, herbs such as *Zinnia acerosa* and *Tiquilia canescens*, as well as shrubs *Baccharis pteronioides* and *Acacia glandulifera*, are common. In these grasslands, the cacti *Turbinicarpus beguinii*, in conservation status by the Mexican government, also grows [68].

We found three exotic species on the study: *Asphodelus fistulosus*, *Malva parviflora*, and *Tribulus terrestris*. The most common in grazed plots is *Asphodelus fistulosus*, a perennial exotic species from Eurasia that grows in areas where rainwater collects. In the region, it is common along the roads, abandoned agricultural fields, and overgrazed areas. The low density of this species may be due to the fact that in the grasslands studied, the climate is characterized by a long dry season and a very cold winter, which, according to [69], act as environmental filters that prevent the establishment of many ruderal and exotic species.

## 5. Conclusions

After five years of cattle grazing exclusion in a native pasture of northern Mexico, no differences in species richness, evenness, or soil nutrients were found. However, some species become more common in ungrazed areas than in grazed plots. In addition, excluding grazing from rangeland benefited the expansion of grasses, whereas forbs increased in the grazed plots, but only for a few species. Thus, the results indicate that the medium-term grazing exclusion did not alter soil nutrient content but enhanced grass growth.

We can expect that differences can increase in the long term. However, we observed some resilience of these areas to grazing, and their evolutionary history explains much of these metanalysis studies' results. In general, exclusion results in a reduction of species richness.

Despite the results, we consider that part of the pastures can be excluded from grazing for longer periods than those in this study (as long as this does not affect the economic performance of the local population) as a way to analyze changes in this natural protected area and to promote an increase in diversity because some species are more linked to excluded areas than others.

**Author Contributions:** Conceptualization, J.R.A., J.A.E.-D. and M.M.; software, J.R.A., J.A.E.-D., E.G. and M.M.; validation, J.R.A., J.A.E.-D., E.G. and M.M.; formal analysis, J.R.A., J.A.E.-D., E.G. and M.M.; investigation, J.R.A. and M.M.; data curation, J.R.A., J.A.E.-D., E.G. and C.G.-M.; writing—original draft

preparation, J.R.A. and J.A.E.-D.; writing—review and editing, C.G.-M., J.A.E.-D., E.G. and M.M.; visualization, J.A.E.-D., E.G. and M.M.; supervision, J.R.A. and J.A.E.-D.; project administration, E.G. and M.M.; funding acquisition, J.R.A., J.A.E.-D., E.G. and M.M. All authors have read and agreed to the published version of the manuscript.

**Funding:** This research received no external funding.

**Data Availability Statement:** Not applicable.

**Acknowledgments:** We wish to thank the staff of the Zapalinamé protected area for supporting this research, especially Sergio C Marines Gómez. We also thank Arturo Cruz-Anaya, Leticia Jiménez, and Rocío Martínez for their assistance during field data collection. Thank you to Jerome Scorer for proofreading and amending the English version of this paper. Many thanks to the Universidad de La Laguna in Tenerife, Spain, for their invaluable support during the preparation of this paper.

**Conflicts of Interest:** The authors declare no conflict of interest.

## Appendix A

**Table A1.** Species family, scientific name, status, functional form, and palatability found in this study.

| Family | Scientific Name | Status | Functional Form | Palatability |
|---|---|---|---|---|
| Euphorbiaceae | *Acalypha monostachya* Cav. | Native | Forb | Non-palatable |
| Euphorbiaceae | *Acalypha phleoides* Cav. | Native | Forb | Non-palatable |
| Poaceae | *Achnatherum eminens* (Cav.) Barkworth | Native | Grasses | Palatable |
| Agavaceae | *Agave asperrima* Jacobi | Native | Shrub | Palatable |
| Nyctaginaceae | *Allionia incarnata* L. | Native | Forb | Non-palatable |
| Amaranthaceae | *Alternanthera repens* (L.) J.F. Gmel. | Native | Forb | Non-palatable |
| Amaranthaceae | *Amaranthus blitoides* S. Watson | Native | Forb | Palatable |
| Amaranthaceae | *Amaranthus hybridus* L. | Native | Forb | Palatable |
| Asteraceae | *Ambrosia confertiflora* DC. | Native | Forb | Non-palatable |
| Malvaceae | *Anoda cristata* (L.) Schltdl. | Native | Forb | Palatable |
| Euphorbiaceae | *Argythamnia neomexicana* Müll. Arg. | Native | Forb | Non-palatable |
| Poaceae | *Aristida adscensionis* L. | Native | Grasses | Non-palatable |
| Poaceae | *Aristida curvifolia* E. Fourn. | Native | Grasses | Non-palatable |
| Poaceae | *Aristida divaricata* Humb. & Bonpl. *ex* Willd. | Native | Grasses | Palatable |
| Poaceae | *Aristida havardii* Vasey | Native | Grasses | Palatable |
| Poaceae | *Aristida pansa* Wooton & Standl. | Native | Grasses | Palatable |
| Poaceae | *Aristida purpurea* Nutt. | Native | Grasses | Palatable |
| Asphodelaceae | *Asphodelus fistulosus* L. | Introduced | Forb | Non-palatable |
| Fabaceae | *Astragalus hypoleucus* S. Schauer | Native | Forb | Non-palatable |
| Asteraceae | *Baccharis pteronioides* DC. | Native | Forb | Non-palatable |
| Asteraceae | *Baccharis salicifolia* (Ruiz & Pav.) Pers. | Native | Forb | Non-palatable |
| Asteraceae | *Bahia absinthifolia* Benth. | Native | Forb | Non-palatable |
| Poaceae | *Bothriochloa barbinodis* (Lag.) Herter | Native | Grasses | Palatable |
| Poaceae | *Bouteloua curtipendula* (Michx.) Torr. | Native | Grasses | Palatable |
| Poaceae | *Bouteloua dactyloides* (Nutt.) J.T. Columbus | Native | Grasses | Palatable |
| Poaceae | *Bouteloua gracilis* (Kunth) Lag. *ex* Griffiths | Native | Grasses | Palatable |
| Poaceae | *Bouteloua hirsuta* Lag. | Native | Grasses | Palatable |
| Rubiaceae | *Bouvardia ternifolia* (Cav.) Schltdl. | Native | Forb | Non-palatable |

**Table A1.** *Cont.*

| Family | Scientific Name | Status | Functional Form | Palatability |
|---|---|---|---|---|
| Poaceae | *Bouteloua uniflora* Vasey | Native | Grasses | Palatable |
| Asteraceae | *Brickellia veronicifolia* (Kunth) A. Gray | Native | Shrub | Non-palatable |
| Buddlejaceae | *Buddleja scordioides* Kunth | Native | Shrub | Palatable |
| Onagraceae | *Calylophus berlandieri* Spach | Native | Forb | Non-palatable |
| Onagraceae | *Calylophus hartwegii* (Benth.) P.H. Raven | Native | Forb | Non-palatable |
| Cyperaceae | *Carex schiedeana* Kuntze | Native | Forb | Palatable |
| Orobanchaceae | *Castilleja sessiliflora* Pursh | Native | Forb | Non-palatable |
| Solanaceae | *Chamaesaracha coniodes* (Moric. *ex* Dunal) Britton | Native | Forb | Non-palatable |
| Asteraceae | *Chaetopappa ericoides* (Torr.) G.L. Nesom | Native | Forb | Non-palatable |
| Amaranthaceae | *Chenopodium foetidum* Lam. | Native | Forb | Non-palatable |
| Rubiaceae | *Clematis drummondii* Torr. & A. Gray | Native | Forb | Non-palatable |
| Cactaceae | *Corynopuntia schottii* (Engelm.) F.M. Knuth | Native | Cacti | Non-palatable |
| Rubiaceae | *Crusea diversifolia* (Kunth) W.R. Anderson | Native | Forb | Non-palatable |
| Boraginaceae | *Cryptantha mexicana* (Brandegee) I.M. Johnst. | Native | Forb | Non-palatable |
| Cucurbitaceae | *Cucurbita foetidissima* Kunth | Native | Forb | Non-palatable |
| Cucurbitaceae | *Cucurbita pepo* L. | Native | Forb | Palatable |
| Cactaceae | *Cylindropuntia imbricata* (Haw.) F.M. Knuth | Native | Cacti | Non-palatable |
| Nyctaginaceae | *Cyphomeris gypsophiloides* (M. Martens & Galeotti) Standl. | Native | Forb | Non-palatable |
| Cyperaceae | *Cyperus niger* Ruiz & Pav. | Native | Forb | Palatable |
| Fabaceae | *Dalea aurea* Nutt. *ex* Pursh | Native | Forb | Palatable |
| Fabaceae | *Dalea bicolor* Humb. & Bonpl. *ex* Willd. | Native | Shrub | Palatable |
| Fabaceae | *Dalea greggii* A. Gray | Native | Shrub | Palatable |
| Fabaceae | *Dalea laniceps* Barneby | Native | Forb | Palatable |
| Fabaceae | *Dalea pogonathera* A. Gray | Native | Forb | Palatable |
| Fabaceae | *Desmanthus painteri* (Britton & Rose) Standl. | Native | Forb | Palatable |
| Convolvulaceae | *Dichondra argentea* Humb. & Bonpl. *ex* Willd. | Native | Forb | Non-palatable |
| Poaceae | *Disakisperma dubium* (Kunth) P.M. Peterson & N. Snow | Native | Grasses | Palatable |
| Caryophyllaceae | *Drymaria anomala* S. Watson | Native | Forb | Non-palatable |
| Asteraceae | *Dyssodia acerosa* DC. | Native | Forb | Non-palatable |
| Acanthaceae | *Dyschoriste linearis* (Torr. & A. Gray) Kuntze | Native | Forb | Non-palatable |
| Asteraceae | *Dyssodia papposa* (Vent.) Hitchc. | Native | Forb | Non-palatable |
| Asteraceae | *Dyssodia pinnata* (Cav.) B.L. Rob. | Native | Forb | Non-palatable |
| Asparagaceae | *Echeandia flavescens* (Schult. & Schult. f.) Cruden | Native | Forb | Non-palatable |
| Cactaceae | *Echinocactus horizonthalonius* Lem. | Native | Cacti | Non-palatable |
| Cactaceae | *Echinocereus pectinatus* (Scheidw.) Engelm. | Native | Cacti | Non-palatable |
| Cactaceae | *Echinocereus reichenbachii* (Terscheck *ex* Walp.) Haage | Native | Cacti | Non-palatable |
| Poaceae | *Elymus elymoides* (Raf.) Swezey | Native | Grasses | Palatable |
| Acanthaceae | *Elytraria imbricata* (Vahl) Pers. | Native | Forb | Non-palatable |
| Poaceae | *Enneapogon desvauxii* P. Beauv. | Native | Grasses | Non-palatable |
| Poaceae | *Erioneuron avenaceum* (Kunth) Tateoka | Native | Grasses | Palatable |

**Table A1.** *Cont.*

| Family | Scientific Name | Status | Functional Form | Palatability |
|---|---|---|---|---|
| Asteraceae | *Erigeron pubescens* Kunth | Native | Forb | Non-palatable |
| Euphorbiaceae | *Euphorbia cinerascens* Engelm. | Native | Forb | Non-palatable |
| Euphorbiaceae | *Euphorbia dentata* Michx. | Native | Forb | Non-palatable |
| Euphorbiaceae | *Euphorbia exstipulata* Engelm. | Native | Forb | Non-palatable |
| Euphorbiaceae | *Euphorbia serrula* Engelm. | Native | Forb | Non-palatable |
| Convolvulaceae | *Evolvulus alsinoides* (L.) L. | Native | Forb | Non-palatable |
| Convolvulaceae | *Evolvulus sericeus* Sw. | Native | Forb | Non-palatable |
| Asteraceae | *Gaillardia pinnatifida* Torr. | Native | Forb | Non-palatable |
| Onagraceae | *Gaura coccinea* Pursh | Native | Forb | Non-palatable |
| Polemoniaceae | *Gilia incisa* Benth. | Native | Forb | Non-palatable |
| Verbenaceae | *Glandularia bipinnatifida* (Nutt.) Nutt. | Native | Forb | Non-palatable |
| Asteraceae | *Gymnosperma glutinosum* (Spreng.) Less. | Native | Shrub | Non-palatable |
| Polygalaceae | *Hebecarpa barbeyana* (Chodat) J.R. Abbot | Native | Forb | Non-palatable |
| Rubiaceae | *Hedyotis nigricans* (Lam.) Fosberg | Native | Forb | Non-palatable |
| Rubiaceae | *Hedyotis rubra* (Cav.) A. Gray | Native | Forb | Non-palatable |
| Fabaceae | *Hoffmannseggia watsonii* (Fisher) Rose | Native | Forb | Palatable |
| Poaceae | *Hopia obtusa* (Kunth) Zuloaga & Morrone | Native | Grasses | Palatable |
| Violaceae | *Hybanthus verbenaceus* (Kunth) Loes. | Native | Forb | Non-palatable |
| Convolvulaceae | *Ipomoea costellata* Torr. | Native | Forb | Non-palatable |
| Convolvulaceae | *Ipomoea purpurea* (L.) Roth | Native | Forb | Palatable |
| Asteraceae | *Laennecia coulteri* (A. Gray) G.L. Nesom | Native | Forb | Non-palatable |
| Polemoniaceae | *Loeselia greggii* S. Watson | Native | Forb | Non-palatable |
| Malvaceae | *Malva parviflora* L. | Introduced | Forb | Palatable |
| Cactaceae | *Mammillaria heyderi* Muehlenpf. | Native | Cacti | Non-palatable |
| Scrophulariaceae | *Mecardonia vandellioides* (Kunth) Pennell | Native | Forb | Non-palatable |
| Oleaceae | *Menodora coulteri* A. Gray | Native | Forb | Palatable |
| Fabaceae | *Mimosa aculeaticarpa* Ortega | Native | Shrub | Palatable |
| Fabaceae | *Mimosa subinermis* (S. Watson) B.L. Turner | Native | Forb | Palatable |
| Nyctaginaceae | *Mirabilis oblongifolia* (A. Gray) Heimerl | Native | Forb | Non-palatable |
| Poaceae | *Muhlenbergia arenicola* Buckley | Native | Grasses | Palatable |
| Poaceae | *Muhlenbergia depauperata* Scribn. | Native | Grasses | Non-palatable |
| Poaceae | *Muhlenbergia phleoides* (Kunth) J.T. Columbus | Native | Grasses | Palatable |
| Poaceae | *Muhlenbergia repens* (J. Presl) Hitchc. | Native | Grasses | Palatable |
| Poaceae | *Muhlenbergia rigida* (Kunth) Kunth | Native | Grasses | Palatable |
| Poaceae | *Muhlenbergia torreyi* (Kunth) Hitchc. *ex* Bush | Native | Grasses | Palatable |
| Poaceae | *Muhlenbergia villiflora* Hitchc. | Native | Grasses | Palatable |
| Poaceae | *Munroa pulchella* (Kunth) L. D. Amarilla | Native | Grasses | Palatable |
| Poaceae | *Nassella leucotricha* (Trin. & Rupr.) R.W. Pohl | Native | Grasses | Palatable |
| Poaceae | *Nassella tenuissima* (Trin.) Barkworth | Native | Grasses | Palatable |
| Brassicaceae | *Nerisyrenia linearifolia* (S. Watson) Greene | Native | Forb | Non-palatable |
| Nostocaceae | *Nostoc commune* Vaucher *ex* Bornet & Flahault | Native | Bacteria | Non-palatable |
| Onagraceae | *Oenothera berlandieri* (Spach) Spach *ex* D. Dietr. | Native | Forb | Non-palatable |

**Table A1.** *Cont.*

| Family | Scientific Name | Status | Functional Form | Palatability |
|---|---|---|---|---|
| Ophioglossaceae | *Ophioglossum engelmannii* Prantl | Native | Fern | Non-palatable |
| Cactaceae | *Opuntia engelmannii* Salm-Dyck | Native | Cacti | Palatable |
| Cactaceae | *Opuntia lindheimeri* Engelm. | Native | Cacti | Palatable |
| Cactaceae | *Opuntia stenopetala* Engelm. | Native | Cacti | Palatable |
| Poaceae | *Panicum hallii* Vasey | Native | Grasses | Palatable |
| Asteraceae | *Parthenium confertum* A. Gray | Native | Forb | Non-palatable |
| Asteraceae | *Parthenium incanum* Kunth | Native | Shrub | Palatable |
| Plantaginaceae | *Penstemon barbatus* (Cav.) Roth | Native | Forb | Non-palatable |
| Montiaceae | *Phemeranthus aurantiacus* (Engelm.) Kiger | Native | Forb | Non-palatable |
| Brassicaceae | *Physaria argyraea* (A. Gray) O'Kane & Al-Shehbaz | Native | Forb | Non-palatable |
| Brassicaceae | *Physaria fendleri* (A. Gray) O'Kane & Al-Shehbaz | Native | Forb | Non-palatable |
| Solanaceae | *Physalis hederifolia* A. Gray | Native | Forb | Non-palatable |
| Phyllanthaceae | *Phyllanthus polygonoides* Nutt. *ex* Spreng. | Native | Forb | Non-palatable |
| Polygalaceae | *Polygala dolichocarpa* S.F. Blake | Native | Forb | Non-palatable |
| Fabaceae | *Pomaria canescens* (Fisher) B.B. Simpson | Native | Forb | Palatable |
| Portulacaceae | *Portulaca pilosa* L. | Native | Forb | Non-palatable |
| Fabaceae | *Prosopis glandulosa* Torr. | Native | Shrub | Palatable |
| Asteraceae | *Pseudognaphalium luteoalbum* (L.) Hilliard & B.L. Burtt | Native | Forb | Non-palatable |
| Asteraceae | *Pseudognaphalium roseum* (Kunth) Anderb. | Native | Forb | Non-palatable |
| Polygalaceae | *Rhinotropis lindheimeri* (A. Gray) J.R. Abbott | Native | Forb | Non-palatable |
| Anacardiaceae | *Rhus microphylla* Engelm. | Native | Shrub | Non-palatable |
| Anacardiaceae | *Rhus virens* Lindh. *ex* A. Gray | Native | Shrub | Non-palatable |
| Fabaceae | *Rhynchosia senna* Gillies *ex* Hook. & Arn. | Native | Forb | Palatable |
| Lamiaceae | *Salvia ballotiflora* Benth. | Native | Shrub | Non-palatable |
| Lamiaceae | *Salvia reflexa* Hornem. | Native | Forb | Palatable |
| Asteraceae | *Sanvitalia ocymoides* DC. | Native | Forb | Non-palatable |
| Apocynaceae | *Sarcostemma crispum* Benth. | Native | Forb | Non-palatable |
| Fabaceae | *Senna demissa* (Rose) H.S. Irwin & Barneby | Native | Forb | Palatable |
| Malvaceae | *Sida abutifolia* Mill. | Native | Forb | Palatable |
| Malvaceae | *Sida spinosa* L. | Native | Forb | Palatable |
| Acanthaceae | *Siphonoglossa pilosella* (Nees) Torr. | Native | Forb | Non-palatable |
| Solanaceae | *Solanum elaeagnifolium* Cav. | Native | Forb | Palatable |
| Malvaceae | *Sphaeralcea angustifolia* (Cav.) G. Don | Native | Forb | Palatable |
| Malvaceae | *Sphaeralcea hastulata* A. Gray | Native | Forb | Palatable |
| Asteraceae | *Stevia tomentosa* Kunth | Native | Forb | Non-palatable |
| Brassicaceae | *Synthlipsis greggii* A. Gray | Native | Forb | Non-palatable |
| Rutaceae | *Thamnosma texana* (A. Gray) Torr. | Native | Forb | Non-palatable |
| Asteraceae | *Thelesperma simplicifolium* (A. Gray) A. Gray | Native | Forb | Non-palatable |
| Asteraceae | *Thymophylla pentachaeta* (DC.) Small | Native | Forb | Non-palatable |
| Asteraceae | *Thymophylla setifolia* Lag. | Native | Forb | Non-palatable |
| Boraginaceae | *Tiquilia canescens* (A. DC.) A.T. Richardson | Native | Forb | Non-palatable |

**Table A1.** *Cont.*

| Family | Scientific Name | Status | Functional Form | Palatability |
|---|---|---|---|---|
| Asteraceae | *Townsendia mexicana* A. Gray | Native | Forb | Non-palatable |
| Zygophyllaceae | *Tribulus terrestris* L. | Introduced | Forb | Non-palatable |
| Cactaceae | *Turbinicarpus beguinii* (N.P. Taylor) Mosco & Zanov. | Native | Cacti | Non-palatable |
| Poaceae | *Urochloa meziana* (Hitchc.) Morrone & Zuloaga | Native | Grasses | Palatable |
| Fabaceae | *Vachellia glandulifera* (S. Watson) Seigler & Ebinger | Native | Shrub | Non-palatable |
| Asteraceae | *Verbesina hypomalaca* B.L. Rob. & Greenm. | Native | Forb | Non-palatable |
| Verbenaceae | *Verbena neomexicana* (A. Gray) Small | Native | Forb | Non-palatable |
| Asteraceae | *Viguiera dentata* (Cav.) Spreng. | Native | Forb | Non-palatable |
| Asteraceae | *Xanthisma spinulosum* (Pursh) D.R. Morgan & R.L. Hartm. | Native | Forb | Non-palatable |
| Asteraceae | *Zinnia acerosa* (DC.) A. Gray | Native | Forb | Non-palatable |

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
