# Peer review of "Changes in Richness and Species Composition after Five Years of Grazing Exclusion in an Endemic Pasture of Northern Mexico"

_land, doi:10.3390/land11111962_

Round 1
Reviewer 1 Report
General comments
The manuscript presents a study on the impact of excluding grazing in an endemic pasture of northern Mexico on plant composition, species richness and evenness. The study was carried out over five years. This is an interesting topic, which falls within the general scope of the Special Issue to which the manuscript was submitted. I am not a native speaker, but I have the impression that the manuscript is written in good English. The title well reflects its contents. The abstract is informative.
The research goals are well formulated. Material and methods are generally well described, but some more information regarding statistical methods is needed (see detailed comments). The results chapter is well written, but the figure legends need improvement (see detailed comments). The discussion is well written. The conclusions are logical and justified.
The list of references contains a large number of 73 positions. The paper is supplemented by one table, six figures and an appendix, all of them important for the manuscript.
Detailed comments
1) Abstract, line20-21: Should be probably “…but not species richness and evenness”.
2) Material and Methods, Study Site, Figure 1: Please add the unit (m) to the size bar.
3) Material and Methods, Design of the experiment, line 93-94: I understand that the authors use the term “control” for the sites where grazing is still active, opposed the sites where grazing is excluded (term “excluded”, i.e. the exclusion of grazing understood as treatment). However, please use these terms consistently throughout the manuscript. For example, in line 160 the term “grazed” is used and in the caption to figure 4 (lines 185-186) both the terms “ungrazed”/”grazed” and “control”/”excluded” are used.
4) Material and Methods, Design of the experiment, line 95: Please write “(10 × 10 m)”.
5) Material and Methods, Design of the experiment, line 104: Maybe, better using “…average annual temperature…”.
6) Material and Methods, Design of the experiment, line 110: The abbreviation ANSM appears for the first time. Please add the full notation.
7) Material and Methods, Design of the experiment, line 117: Maybe, better using “…extractable phosphorus using the Olsen method…”.
8) Material and Methods, Statistical analyses, Ordination techniques: The authors use both linear (PCA) and unimodal (DCA) ordination methods. According to literature (e.g. Lepš & Šmilauer, 2003) the selection of the appropriate model depends on the first gradient of a DCA (or DCCA in the case of constraint analyses). If this was applied by the authors, please mention in the manuscript.
It is also not clear if the analysis of species composition between control and excluded plots was done by PCA (line 130) or by DCA (line 135). In the results a DCA diagram is shown.
9) Results, caption to table 1, lines 150-152: Not all of the abbreviations in the topline of the table are clear. I assume that “MO” means “organic matter content”, “P Ols” means “phosphorus Olsen” and “TN” means “soil total nitrogen”. I propose to ad these abbreviations to the methods (lines 113-119). However, Fe, Mn, B, S are not mentioned in the methods. On the other hand, in the methods are mentioned “electrical conductivity”, “Cation Exchange Capacity” and “Qualitative N, P, K”, which are not shown in the table. Moreover, some of the abbreviations are not used consistently: for example, in figure 5 is used “Polsen” and “OM”. In my opinion the abbreviations should be used consistently.
10) Results, line 154-160: Is it true that the values for grasses were higher in the control plots? With respect to forbs cover the values were also higher in the control plots, but in figure 3a and 3b the line type for the higher values is different (see also next comment). However, in the discussion (lines 228-229) is written that for grasses the higher values were in the excluded plots.
11) Results, caption to figure 3: Please indicate which line type denotes “control” and which denotes “ excluded”.
12) Discussion, line 229: Should be “…grasses in the excluded plots…”.
13) Discussion, line 232: Should be “…that this stratum has the ability…”.
14) Discussion, line 283: “The ordination analyses revealed…”: which analysis – PCA or DCA? The use of the plural “analyses” implies that both. Please specify (see also comment 8).
15) Discussion, lines 306-307: The grammar of the sentence seems to be faulty. Please check.
Conclusion
The authors present a paper, which is interesting and scientifically important. Generally, the paper is well written, traceable and understandable. Nevertheless, I have quite some comments where I see necessity for improvement. However, since these comments concern generally more formal aspects which should be rather easy to be solved by the authors, I propose minor revision of the manuscript before publication.
Author Response
REFEREE I
Comment: The manuscript presents a study on the impact of excluding grazing in an endemic pasture of northern Mexico on plant composition, species richness and evenness. The study was carried out over five years. This is an interesting topic, which falls within the general scope of the Special Issue to which the manuscript was submitted. I am not a native speaker, but I have the impression that the manuscript is written in good English. The title well reflects its contents. The abstract is informative.
The research goals are well formulated. Material and methods are generally well described, but some more information regarding statistical methods is needed (see detailed comments). The results chapter is well written, but the figure legends need improvement (see detailed comments). The discussion is well written. The conclusions are logical and justified.
The list of references contains a large number of 73 positions. The paper is supplemented by one table, six figures and an appendix, all of them important for the manuscript.
Answer: We really thanks the effort and time of the referee, as well as the positive comments of the authors about our study.
Comments:
Detailed comments
1) Abstract, line20-21: Should be probably “…but not species richness and evenness”.
2) Material and Methods, Study Site, Figure 1: Please add the unit (m) to the size bar.
3) Material and Methods, Design of the experiment, line 93-94: I understand that the authors use the term “control” for the sites where grazing is still active, opposed the sites where grazing is excluded (term “excluded”, i.e. the exclusion of grazing understood as treatment). However, please use these terms consistently throughout the manuscript. For example, in line 160 the term “grazed” is used and in the caption to figure 4 (lines 185-186) both the terms “ungrazed”/”grazed” and “control”/”excluded” are used.
4) Material and Methods, Design of the experiment, line 95: Please write “(10 × 10 m)”.
5) Material and Methods, Design of the experiment, line 104: Maybe, better using “…average annual temperature…”.
6) Material and Methods, Design of the experiment, line 110: The abbreviation ANSM appears for the first time. Please add the full notation.
7) Material and Methods, Design of the experiment, line 117: Maybe, better using “…extractable phosphorus using the Olsen method…”.
Answer: We have corrected and implemented all the changes suggested by the referee. We also standardized the use of contro/grazed along the manuscript.
Comments: 8) Material and Methods, Statistical analyses, Ordination techniques: The authors use both linear (PCA) and unimodal (DCA) ordination methods. According to literature (e.g. Lepš & Šmilauer, 2003) the selection of the appropriate model depends on the first gradient of a DCA (or DCCA in the case of constraint analyses). If this was applied by the authors, please mention in the manuscript.
It is also not clear if the analysis of species composition between control and excluded plots was done by PCA (line 130) or by DCA (line 135). In the results a DCA diagram is shown.
Answer: We included some information on the main text about the linear gradient of soil characteristics and the unimodal gradient of species composition. Because of that we use PCA for soil and DCa for species composition.
Comments: 9) Results, caption to table 1, lines 150-152: Not all of the abbreviations in the topline of the table are clear. I assume that “MO” means “organic matter content”, “P Ols” means “phosphorus Olsen” and “TN” means “soil total nitrogen”. I propose to ad these abbreviations to the methods (lines 113-119). However, Fe, Mn, B, S are not mentioned in the methods. On the other hand, in the methods are mentioned “electrical conductivity”, “Cation Exchange Capacity” and “Qualitative N, P, K”, which are not shown in the table. Moreover, some of the abbreviations are not used consistently: for example, in figure 5 is used “Polsen” and “OM”. In my opinion the abbreviations should be used consistently.
Answer: We have cofrrected all the problems and missed information in this part as suggested by the referee.
Comments: 10) Results, line 154-160: Is it true that the values for grasses were higher in the control plots? With respect to forbs cover the values were also higher in the control plots, but in figure 3a and 3b the line type for the higher values is different (see also next comment). However, in the discussion (lines 228-229) is written that for grasses the higher values were in the excluded plots.
Answer: The referee is right and there was a confusion in the explanation of the results.
Comments: 11) Results, caption to figure 3: Please indicate which line type denotes “control” and which denotes “ excluded”.
12) Discussion, line 229: Should be “…grasses in the excluded plots…”.
13) Discussion, line 232: Should be “…that this stratum has the ability…”.
14) Discussion, line 283: “The ordination analyses revealed…”: which analysis – PCA or DCA? The use of the plural “analyses” implies that both. Please specify (see also comment 8).
15) Discussion, lines 306-307: The grammar of the sentence seems to be faulty. Please check.
Answer: All of these comments have been implemented.
Reviewer 2 Report
This study is a contribution to the very important knowledge about the grazing influence on the vegetation. Historical grazing has influenced the species composition and it is important to know how grazing exclusion affects the vegetation. This topic has been involved in numerous studies, but information from different bioclimatic regions is of particular importance. Current study was conducted in semiarid pastures which perform specificity due to the dry climatic conditions and such data is still scarce. The study design is well organized. Considering the livestock number and pasture area (Lines 80-81), the grazing intensity is not high. It would be of benefit for the study if the authors could comment the maximum load for cattle and horses on the unit area.
Methods: I am wondering how the authors have decided the cover estimate in the case of threshold values?
Results: I would recommend to shift the results about nutrient composition (lines 188-200) below the Table 1 because they better explain the differences in nutrient content. Further on you may continue with the species data. You could omit some species labels in DCA diagram by defining higher threshold for abundance and this will make Figure 6 better for reading. Also you mention that the introduced species Asphodelus fistulosus, Malva parviflora, and Tribulus terrestris express similar quantity in both plots. But Tribulus terrestris appears to be more frequent in the control plots. It is an interesting result that forbs appear to be better adapted to cattle grazing than grasses.
Discussion: Probably the main reason for the lack of difference is low grazing intensity (lines 265-268; 278-279). The statement in lines 280-282 is not supported by any data in the results. The authors state that three Bouteloua species are presented with high coverage and Bouteloua dactyloides is the dominant. How this species is a dominant but without high cover? Please rewrite the sentence between lines 299-303 because now it is not easy to understand. Do you consider exotic plants as alien plants which could be harmful to the native communities?
Conclusions: I suggest to reconsider the conclusions. They should include the main findings and generally avoid citations. The resilience of vegetation to grazing could be commented by the grazing legacy. “Strong relationship with weather variability” was not proven. Suggestion for further management is always of use and it will be good to precise the length of exclusion. This will help the local authorities to govern in the best way the protected area.
Minor remarks:
Line 70: Please, describe in full the abbreviation BSKw
Fig.1 – The figure does not show both sampling plots. It is sufficient to mention in the figure caption “Study site showing the sampling plots located in the Coahuila State, Mexico”. Federal roads and state roads are not well distinguished. Please, add unit (m) on the map reference scale.
Lines 90-94 – Please, mention how far from each other are excluded from grazed plots.
Table 1- What is TN?
Figure 3 – Please, include in the legend the meaning of solid line and dashed line.
line 232 – ‘de’ should be ‘the’
Line 267 – in the discussion functional types analysis is mentioned, please specify in the methods that you consider grasses, forbs and woody plants as functional types.
Author Response
Comment: This study is a contribution to the very important knowledge about the grazing influence on the vegetation. Historical grazing has influenced the species composition and it is important to know how grazing exclusion affects the vegetation. This topic has been involved in numerous studies, but information from different bioclimatic regions is of particular importance. Current study was conducted in semiarid pastures which perform specificity due to the dry climatic conditions and such data is still scarce. The study design is well organized. Considering the livestock number and pasture area (Lines 80-81), the grazing intensity is not high. It would be of benefit for the study if the authors could comment the maximum load for cattle and horses on the unit area.
Answer: We thank the reviewer for the time dedicated to the manuscript. Unfortunately, we only have average information of the animal’s units per area, but base in personal communication of managers of the area is relatively constant along the productive vegetation period of the year.
Comment: Methods: I am wondering how the authors have decided the cover estimate in the case of threshold values?
Answer: We estimated visually the species cover following the van der Maarel (2007) propose. As long as it is a traditional and robust method, we didn’t include the reference.
Comment: I would recommend to shift the results about nutrient composition (lines 188-200) below the Table 1 because they better explain the differences in nutrient content. Further on you may continue with the species data.
Answer: The location of the table is due to the comments about the environmental characteristics, that is included in table 1. We didn´t discuss about nutrients specifically, the comments about nutrient composition are in below the PCA figure. We are centered in the discussion of nutrient composition, more that nutrients specifically.
Comments: You could omit some species labels in DCA diagram by defining higher threshold for abundance and this will make Figure 6 better for reading.
Answer: We consider that one of the results of this study is to represent the high species richness. So we have try to keep all the species in the DCA graph, as long as we made the effort to make it visually feasible.
Comment: Also you mention that the introduced species Asphodelus fistulosus, Malva parviflora, and Tribulus terrestris express similar quantity in both plots. But Tribulus terrestris appears to be more frequent in the control plots. It is an interesting result that forbs appear to be better adapted to cattle grazing than grasses.
Answer: You are right, but base in our ISI analysis, differences in presence can not be consider significant. We only have a restricted number of species that presented significant differences following the statistical method. Unfortunately, no one of the exotic species.
Comments: Discussion: Probably the main reason for the lack of difference is low grazing intensity (lines 265-268; 278-279).
Anwwer: Yes we agree, we included that time a low grazing pressure can have an impact on the results.
Comments: The statement in lines 280-282 is not supported by any data in the results.
Answer: We have removed that general stamen.
Comments: The authors state that three Bouteloua species are presented with high coverage and Bouteloua dactyloides is the dominant. How this species is a dominant but without high cover? Please rewrite the sentence between lines 299-303 because now it is not easy to understand.
Answer: We agree with the referee, this were more a general comment, and we have removed that paragraph and references.
Discussion: Do you consider exotic plants as alien plants which could be harmful to the native communities?
Answer: Base at nowadays presence and abundance with consider that are no altering ecological process for now.
Comments: Conclusions: I suggest to reconsider the conclusions. They should include the main findings and generally avoid citations. The resilience of vegetation to grazing could be commented by the grazing legacy. “Strong relationship with weather variability” was not proven. Suggestion for further management is always of use and it will be good to precise the length of exclusion. This will help the local authorities to govern in the best way the protected area.
Answer: We have made some modification of the section avoiding some comments and recommended by the referee.
Comments: Minor remarks:
Line 70: Please, describe in full the abbreviation BSKw
Fig.1 – The figure does not show both sampling plots. It is sufficient to mention in the figure caption “Study site showing the sampling plots located in the Coahuila State, Mexico”. Federal roads and state roads are not well distinguished. Please, add unit (m) on the map reference scale.
Lines 90-94 – Please, mention how far from each other are excluded from grazed plots.
Table 1- What is TN?
Figure 3 – Please, include in the legend the meaning of solid line and dashed line.
line 232 – ‘de’ should be ‘the’
Line 267 – in the discussion functional types analysis is mentioned, please specify in the methods that you consider grasses, forbs and woody plants as functional types.
Answer: All of these minor changes have been implemented in the main text.
We thank the referee for the time and effort dedicate to the manuscript that we consider improved the quality significatively.